# Literature Review and Content Analysis of Bullying Assessments: Are We Measuring What We Intend to?

**DOI:** 10.3390/ijerph22010029

**Published:** 2024-12-29

**Authors:** Katherine A. Graves, Lindsey G. Mirielli, Cannon Ousley, Chad A. Rose

**Affiliations:** 1Department of Teacher Assessment and Preparation, University of Texas at Arlington, Arlington, TX 75050, USA; 2Juniper Gardens, Kansas University, Kansas City, MO 66101, USA; lmirielli@ku.edu; 3Department of Special Education, University of Missouri-Columbia, Columbia, MO 65201, USA; cort6@umsystem.edu (C.O.); rosech@missouri.edu (C.A.R.)

**Keywords:** bullying, assessment, victimization

## Abstract

The initial phase in any initiative aimed at preventing bullying involves evaluating the present prevalence to pinpoint students who might be more susceptible to involvement in the bullying dynamic. Assessment serves as a guide for shaping future decisions regarding intervention. The purpose of this study was to identify and evaluate current assessment tools to determine the extent to which the bullying dynamic is currently measured. The results indicated that assessment tools measured verbal bullying/victimization most frequently, followed by relational and physical. Also, items measured repetition and intent about 50% of the time, while they measured power imbalance less frequently (i.e., 25%). The importance of matching an appropriate assessment to a school’s needs is emphasized. Implications for both researchers and practitioners are discussed.

## 1. Literature Review and Content Analysis of Bullying Assessments: Are We Measuring What We Intend to?

Bullying involvement is a common concern for school professionals, parents, and students. Currently, one in five students in K-12 settings experiences bullying each year [1]. Negative outcomes associated with bullying involvement include psychosomatic symptoms [2,3], lower academic achievement [4,5], higher levels of externalizing and internalizing behaviors [6], increased mental health problems [7,8], and suicidal ideation [9]. Due to the gravity of these adverse outcomes, schools have been tasked with implementing school-wide systems focused on the prevention of and intervention with bullying. A crucial first step in any bullying prevention effort is the assessment of current rates of bullying to identify students who may be at increased risk of bullying involvement and to guide future intervention decisions [10]. While various assessment tools are available, minimal research has been conducted on the content of individual items within these tools. Therefore, this study aimed to examine the extent to which bullying assessment tools address the constructs outlined in the definition of bullying and the various characteristics of the bullying dynamic (e.g., roles, types).

### 1.1. Bullying Defined

Bullying is defined as having three constructs: (1) unwanted aggressive behavior exhibited by another youth or group of youths (not siblings or partners), (2) a perceived power imbalance (e.g., a child who is perceived as more popular or stronger aggresses towards a child that is perceived as less popular or weaker), and (3) repetition across time or victims [11]. Furthermore, bullying can be experienced directly or indirectly [12]. Direct bullying occurs in the presence of the targeted individual (e.g., a physical altercation or directed aggressive verbal comments). Indirect bullying occurs when the individual is absent (e.g., spreading rumors or cyberbullying). It is important to note that not every form of aggression is bullying. For example, instrumental aggression, where the aggression appears necessary to protect oneself or others, or retaliatory aggression, where the aggression of another individual serves as the antecedent for aggressive behavior, is not considered bullying [13]. Additionally, jostling (i.e., play fighting) is not considered bullying as there is no perceived power imbalance, and it may not be repeated over time [14]. Given the complex nature of bullying, an increased understanding of assessment tools—an understanding that considers each construct of the bullying definition—is imperative.

### 1.2. Types and Roles

Bullying is a pervasive issue that can manifest in various forms, each with distinct characteristics that contribute to the complexity of the issue. Currently, six different types of bullying are recognized: verbal, physical, relational (e.g., spreading rumors, exclusion from social groups), cyber (i.e., online), damage to property, and sexual (e.g., making unwanted sexual comments to others, Children’s Scale of Hostility and Aggression: Reactive/Proactive) [14,15]. These types of bullying vary within schools. For example, in a sample of 24,620 high school students, [16] found that 8% of girls and 9.5% of boys experienced physical bullying. About 9% of boys experienced relational bullying, whereas 15.6% of girls experienced relational. Further, 22% of girls and 16.9% of boys endorsed a verbal form of bullying victimization. Lastly, 7.5% of girls and 4.3% of boys reported experiencing cyberbullying [16]. Not only do rates vary by the type of bullying, but they also vary by gender [17].

An additional layer of intricacy in comprehending bullying lies in individuals’ diverse roles, specifically, a student who engages in bullying behavior—or “bully,” a student who is the recipient of bullying behavior—or victim, and bystanders. Bystanders are individuals who do not play a primary role but are present during the bullying incident. Currently, there are four accepted types of bystanders: (1) kids who assist (e.g., kids who encourage the bully or join in), (2) kids who reinforce (e.g., kids who are the audience and may laugh or cheer on the perpetrator), (3) kids who defend (e.g., kids who defend the victim), and (4) outsiders (e.g., kids who are the audience and neither reinforce nor defend) [14,18]. Research on bystanders has found that peers only intervene 10% of the time despite being present 85% of the time [19]. The rates of roles vary with the school, time, and context. For example, a meta-analysis found that there is a mean prevalence of 35% of middle school and high school students involved in traditional bullying dynamics (i.e., those who engage in bullying behavior and those who are victimized) [20]. Differentiating between the two, several large-scale studies found that approximately 4–9% of students often engage in bullying behaviors, and 9–25% of students are bullied [21].

### 1.3. Current Research on Assessment Tools

Schools must find an accurate way to measure bullying involvement before implementing interventions and systems to reduce it. *Measuring Bullying Victimization, Perpetration, and Bystander Experiences: A Compendium of Assessment Tools* [22] is a resource that provides researchers, educators, and policymakers with validated instruments for assessing the multifaceted nature of bullying. It includes 33 measures and, more specifically, characteristics of the assessment tool, target groups, psychometrics, the developer, measure items, response categories, and the information provided to respondents at the beginning of the measure. An assessment was included in the compendium if the assessment referred to the construct of “bullying”, even if the authors did not assess the power differential and chronicity of the target behavior and did not label the behavior as bullying for the research participants. The assessment had to assess constructs related to bullying, such as physical aggression, relational aggression, sexualized and homophobic bullying, and bystander experiences. Further, the assessment had to have been administered to respondents between 12 and 20 years of age. Since the bulk of work on bullying began in the 1980s, the tools had to be developed or revised between 1980 and 2007 (i.e., when the literature review was concluded). Finally, when available, tools had to be self-administered in English and published in a peer-reviewed journal or book, including psychometric information about the assessment tool. This compendium offers tools for evaluating the experiences of victims, perpetrators, and bystanders, enabling a nuanced understanding of bullying dynamics across diverse settings.

Much of the research on bullying assessment tools has focused on the structure of these assessments. For example, when measuring bullying, multi-item scales should be used to give a more valid, accurate, and reliable measurement [23,24,25]. Single items often do not represent complex issues well, may lack precision, and are prone to a high degree of random error [25,26].

Research has also focused on the language used within these tools. In a systematic review of assessment tools, [27] found that most assessments employed various terminology for measuring bullying. Of the assessments analyzed, 11 provided a specific definition of bullying, while 13 incorporated “bullying” within their assessment tools. The directions for persons using the assessment tool also provide differing information on when the bullying has occurred. For instance, one assessment directs, “Think about what happened DURING THE LAST 7 DAYS, when you answer these questions”, [28] and another directs, “Choose how many times you did this activity or task in the last 30 days. In the last 30 days” [29]. Consequently, the results underscore a notable lack of consistency in measurement approaches across various constructs, complicating the comparison of prevalence rates between assessment tools.

Further, researchers have investigated the most common reporting methods of assessments measuring bullying involvement. Self-report assessments are the most widely used [6,30,31,32]; other reporting options include peer reports, parent reports, teacher reports, and observations [33]. Most studies involve only one type of report, and the use of multiple reporters could be advantageous in reducing possible bias [25,33]. One review and content analysis of assessments [27] found that student self-reporting was the primary reporting method, not allowing multiple perspectives to be assessed.

Researchers have systematically reviewed the psychometric properties of available assessments. [32] found that six measurement papers demonstrated a quality score of 75% or above, indicating that there is limited evidence to support the reliability, validity, and responsiveness of existing youth bullying assessments [34]. While research surrounding the structure, reporting methods, and psychometric properties of bullying assessment tools is widely available, minimal research has focused specifically on the content of individual assessment tools and the extent to which the definition, types, and roles of bullying are measured.

### 1.4. Purpose

This study aimed to evaluate assessment tools to determine the extent to which bullying is currently measured. The following research questions were answered to fulfill the aforementioned purpose:-To what extent do assessment tools of bullying involvement measure the three constructs of the definition of bullying (i.e., intent, imbalance, repetition)?-To what extent do assessment tools of bullying involvement measure the different characteristics of the bullying dynamic (i.e., type, role)?

## 2. Methods

### 2.1. Article Selection Procedures

#### Search Procedures

Prior to conducting our search, we began by reviewing the *Measuring Bullying Victimization, Perpetration, and Bystander Experiences: A Compendium of Assessment Tools* [22]. All 33 assessments and their references were included in the literature collection. Then, we conducted an extensive electronic search using three databases (i.e., Academic Search Premier, APA PsychInfo, and Education Resources Information Center [ERIC]) to identify articles from 1980 to 2022 using the search terms “bully* and measure*” and “bully* and scale*”. Because the *Compendium of Assessment Tools* [22] included assessment tools published from 1990 to 2007, we selected articles published during the years 1980–2022 (i.e., the end date of the literature search). We updated the search beginning in 1980 to ensure no assessments were missed in the 10 years before the publication of the *Compendium of Assessment Tools* [22].

The initial search resulted in 22,889 articles. First, the abstracts and titles were screened. If the article indicated that the assessment tool addressed constructs related to bullying and was administered to respondents between 12 and 20 years of age and had not already been identified in the *Compendium of Assessment Tools*, it was retained. Given the number of search results, a discontinuation criterion was put in place, and the search ceased after 1000 consecutive studies that did not meet the inclusion criteria.

### 2.2. Inclusion and Exclusion Criteria

To be included, the article had to meet the following criteria: (a) include information about the development of the assessment and Cronbach’s alpha of the assessment tool; (b) must claim to assess bullying or constructs related to bullying (e.g., physical aggression, relational aggression, sexualized and homophobic bullying, and bystander experiences); (c) the assessment must have been administered to respondents between 12 and 20 years of age; (d) the assessments had to be developed or revised between 1980 and 2022 (i.e., when the review of the literature was concluded); and (e) the assessment tool must be in English. Articles were excluded if they (a) did not include information about the development of the assessment (e.g., an article describing a research study using the assessment tool as outcome data), (b) the assessment did not claim to assess bullying or constructs related to bullying (e.g., assessment measuring solely the school climate broadly), (c) were administered to respondents younger than 12 years of age or older than 20 years of age, (d) the assessment was developed before 1980 or after 2022, and (e) the assessment was not available in English.

Information for each article that met the criteria for inclusion was compiled in a spreadsheet, including the article citation, name of assessment, purpose of the study, and Cronbach’s alpha. Duplicates from the *Compendium of Assessment Tools* [22] were removed. From the literature search, 15 assessment tools were included in the final analysis, resulting in 48 total assessment tools for the content analysis.

### 2.3. Content Analysis

For the analysis, each item from the assessment tools was coded across four domains: definition, type, role, and other (e.g., demographic item or item related to a construct outside of bullying involvement). Prior to analysis, each assessment tool was given a unique identifier (i.e., 1–48). Then, each item from the assessment tool was copied into an individual row in an Excel document that included (1) the assessment tool’s unique identifier, (2) the individual item number, (3) the stem of the item (e.g., “In the previous 30 days, how often have you...”), (4) the item (e.g., “been hit, kicked, or pushed by someone at your school”), (5) the response options, and (6) any other information (e.g., if the authors of the assessment tool indicated a construct the item fell into). Following the creation of the Excel document, each item was coded dichotomously (i.e., 1 for yes or 0 for no) across the four domains described below.

### 2.4. Definition

Each item was coded according to [11]’s definition of bullying. This domain required the coder to identify whether the item appropriately addressed intent, repetition, and/or power imbalance. Each aspect of the definition was coded dichotomously (i.e., 1 for yes or 0 for no). Throughout this process, decision rules were created. Items that used phrases such as “to another student” (e.g., “I made sexual jokes, comments, or gestures to another student(s)”) or “on purpose” (e.g., “I kept another student(s) out of things on purpose, excluded him or her from my group of friends or completely ignored him or her”) were automatically coded as meeting the criteria for intent. When coding for repetition, coders referred to the response choices (e.g., 0 times, 1–2 times, 3–5 times, 6+ times) and/or stem. If the item allowed the respondent to indicate an act occurred more than once, then repetition was selected for the item. Power imbalance was the most difficult to determine, as items had to indicate clear physical, symbolic, economic, informational, cultural, or social capital [14]. Further, items were coded as addressing power imbalance if there was a mention of a protected class (i.e., disability) or a group of students aggressing towards a single student [14]. Finally, if the item was determined not to assess intent, repetition, or power imbalance, a one was placed in the “none” category.

### 2.5. Type, Role, and Additional Information

In addition to coding for the definition of bullying, each item was coded for the type of bullying/victimization, role within the bullying dynamic, and additional information. These categories were also coded dichotomously (i.e., 1 for yes or 0 for no). Each item could be coded into one type of bullying/victimization as outlined by the Health Resources and Services Administration [14]: physical, verbal, relational, sexual, cyber, illegal, destruction to property, and none. An additional category of “General Bullying” was created by the lead researcher. This category was selected when the item addressed bullying behaviors but did not specify the type listed above. Decision rules for this coding category were also created. For example, verbal bullying was selected if an item referred to gestures, stares, written, and teasing [14].

Each item was coded for the role it targeted within the bullying dynamic. These roles include bully, victim, bystander-assist, bystander-defender, bystander-reinforcer, bystander-outsider, and none [14]. Finally, individual items were coded for any additional information present. This included a spot to indicate if the item was asking for demographic information or was assessing a construct other than bullying involvement (e.g., prosocial behavior, academics, impulsivity). If an item was coded within the “additional information” category, all other domains (i.e., definition, type, role) were coded as “none”.

Additional decision rules were made that had an impact across multiple domains. For example, definitionally, physical bullying [35], cyberbullying [36], and teasing (i.e., verbal bullying) [37] all include intent by the perpetrator. Therefore, if an individual item was coded as any of these three types, they were also coded as addressing intent in the definition domain. The definition of relational bullying automatically includes power imbalance and intent [14]; therefore, if an item was coded as relational bullying, the item was automatically coded as addressing these two categories in the definition domain. Items that did not address any of the types of bullying/victimization were marked with a 1 in the “none” category. Following the completion of coding individual items, the overall assessment tools were reviewed; if the assessment tool did not include at least one item that measured a type or role, it was excluded from the final analysis (*n* = 1).

### 2.6. Interrater Reliability

To ensure the validity and reliability of the coding, assessments were coded by the first two authors for 45% of the articles (*n* = 21), resulting in an interrater reliability agreement of 100%. First, the last author trained the coders in the domain definitions. The number of items agreed on divided by the total number of items coded resulted in a percentage of agreement. Following coding for the first three articles, the reliability before agreement was 73%. The authors continued this process seven times in order to conduct reliability for 45% of the articles. The remaining assessment tools (*n* = 26) were divided in half and coded. The final interrater reliability agreement was 100%.

## 3. Results

The 48 assessment tools were coded and descriptively analyzed to answer the proposed research question. Results are presented by individual items and scales that fall into the following categories: (a) bully-only scales, (b) victim-only scales, (c) bully and victim scales, and (d) bully, victim, and bystander scales.

### 3.1. Descriptive Information

Assessment tools can be created for teachers, parents, or student (i.e., self or peer) reports. Of the 47 assessments, 43 were student reports, 3 were parent reports, and 1 was intended for teachers to report on bullying. A few assessments had multiple versions for different reporters (e.g., student and teacher). In these cases, the student-self-report assessments were used for analysis. The age range for the assessment tools is 3 to 24 years old. While inclusion criteria specified grades K-12, there is a parent-report assessment tool (i.e., Children’s Scale of Hostility and Aggression: Reactive/Proactive [C-SHARP]) that can be used for ages 3–21, and the Perception of Teasing Scale (POTS) targets students aged 17–24. The mode of the age range for the assessment tools is 10–15 (i.e., approximately 5th–9th grades). The average number of items was 32, ranging from 4 to 135 items per assessment tool. Items were organized into constructs with an average of 3 constructs per assessment and a range from 1 to 10 constructs.

### 3.2. Item-Level Analysis

The frequencies and percentages for the item-level analysis were calculated using Excel. Of the 1499 total items, victimization was most commonly measured, followed by bullying and bystander behavior. Additionally, verbal bullying/victimization was measured most frequently, followed by relational, physical, cyberbullying, and sexual. The fewest number of items measured property damage and illegal behavior. Further, approximately half of the 1499 items measured repetition and intent. Only a quarter of the items measured power imbalance. Finally, 23% (*n* = 347) of the items measured all three constructs. See Table 1 for the frequencies and percentages of items within each coded category.

Following item analysis, each scale was placed into one of four categories based on which roles they measured: bully only, victim only, bully and victim, and bully, victim, and bystander. The following section includes assessment-level analyses and reports on the type of bullying involvement (i.e., physical, verbal, relational, damage to property, sexual, and cyber) and definition components (i.e., intent, repetition, and power imbalance). 

### 3.3. Bully-Only Assessments

Four total assessments and 103 items exclusively measured bullying perpetration from the sample (Table 2). The total number of items per assessment ranged from 9 to 58 items. Physical bullying was most frequently measured by the assessments (100%, *n* = 4), then verbal (75%, *n* = 3), relational (75%, *n* = 3), damage to property (25%, *n* = 1), sexual bullying, (25%, *n* = 1), and cyberbullying (0%, *n* = 0), and all four of the assessments had items that measured none of the types. Further, every assessment had items that measured intent (100%, *n* = 4), and three assessments had items that measured repetition and power imbalance. Out of all four assessments, there were zero where every item measured repetition, intent, and/or power imbalance.

### 3.4. Victim-Only Assessments

Twelve total assessments were categorized into the victim-only group. The total number of items per assessment measured ranged from 5 to 133 (Table 3). Verbal victimization was the most frequent type of victimization measured (100%, *n* = 12), then relational (83%, *n* = 10), physical (83%, *n* = 10), damage to property (67%, *n* = 8), cyber victimization (42%, *n* = 5), and sexual victimization (25%, *n* = 3), and 33% (*n* = 4) of the assessments included items that did not measure any of the types of victimization. All 12 assessments had items that measured the three domains of the definition. There were five assessments in which every item measured repetition and intent. However, the percentage of items per assessment measuring power imbalance ranged from 11% to 55%; the assessments measured power imbalance less than repetition or intent.

### 3.5. Bully and Victim Assessments

The bully and victim assessment category included 20 assessments, with total items per assessment ranging from 8 to 42 (Table 4). Again, verbal bullying and victimization were measured the most frequently (80%, *n* = 16), followed by physical (75%, *n* = 15), relational (65%, *n* = 14), cyberbullying and victimization (30%, *n* = 6), damage to property (30%, *n* = 6), and sexual bullying and victimization (15%, *n* = 3), and 50% (*n* = 10) of the assessments included items that did not measure any bullying or victimization. Every assessment had items that measured intent; however, only 80% of the assessments measured repetition, and 85% (*n* = 17) measured power imbalance. Similar to the victim-only assessment category, there were five assessments in which every item measured repetition and intent. While 85% of the assessments measured power imbalance, there was one assessment in which only 1 item out of the total 28 items had language to include an imbalance of power.

### 3.6. Bully, Victim, and Bystander Assessments

Assessments were included in this category if they measured bullying and victimization and included items that measured bystanders who had witnessed bullying (Table 5). Twelve total assessments were included, with the number of items per assessment ranging from 5 to 135. Like the above categories, most assessments measured verbal bullying and victimization (75%, *n* = 9) followed by physical (67%, *n* = 8), relational (58%, *n* = 7), cyber (25%, *n* = 3), damage to property (17%, *n* = 2), sexual (8%, *n* = 1), and illegal (8%, *n* = 1), and 58% of the assessments included items that did not measure any bullying, victimization, or bystander involvement. All but one of the assessments measured intent, and all but two measured repetition and power imbalance. This category only included one assessment in which every item measured repetition and intent. While 83% of the assessments measured power imbalance, many of the assessments, especially those with a high number of items, had very few items that measured an imbalance of power (e.g., an assessment that included 53 items had only 1 item that measured an imbalance of power).

## 4. Discussion

Bullying remains a common concern for school-aged youth, given the detrimental outcomes associated with bullying involvement [2,3,4,5,6,7,8,9]. The first step in implementing bullying prevention efforts is the assessment of bullying prevalence [10]. Currently, the extant literature on bullying assessment tools primarily focuses on the psychometric evaluations of assessments [34], language used [27], structure [25], and types of reporting methods [25]. Previously, resources have been created for educators compiling available assessment tools, including [22]’s *Measuring Bullying Victimization, Perpetration, and Bystander Experiences: A Compendium of Assessment Tools*. The current study extends the prior literature surrounding bullying assessment tools by identifying additional assessment tools not included in the *Compendium of Assessment Tools* and determining the extent to which each tool addresses the three constructs of the definition of bullying, as well as other key characteristics of the bullying dynamic (i.e., types, role).

The bullying assessment tools analyzed in this study are designed to be administered by schools to understand student levels of bullying involvement and are the first step in informing prevention and intervention. It is crucial that schools use assessment tools that are reliable and valid and measure what they intend to do. According to the [14] definition of bullying, bullying or victimization must include three domains: repetition, intent, and power imbalance. Results of this study indicate that only 23% (*n* = 339) of the total items measure all three definition domains. These results are similar to previous findings in the literature, where out of a sample of 135 victimized students, labeled victims by the Revised Olweus’ Bully/Victim Questionnaire (BVQ), only 43.1% reported that their victimization included repetition, intent, and power imbalance [80]. Results from this study, paired with [80]’s findings, suggest that students could be categorized as experiencing victimization without truly experiencing all three domains of the bullying definition.

The results of this study were presented in the following categories of assessment tools: bully-only; victim-only; bully and victim; and bully, victim, and bystander. The purpose of this presentation was to align with the literature suggesting that roles in the bullying dynamic vary based on time and context [81]. To illustrate, a student may engage in bullying behaviors at recess yet experience victimization during reading, resulting in the same student being characterized as both a bully and a victim, depending on the time and context. Given the complex nature of the bullying dynamic, schools should seek assessment tools that measure each role of the bullying dynamic to capture the students who could be categorized as bully-victims [70], as opposed to assessment tools that solely focus on one role. Therefore, using a comprehensive assessment tool that addresses all definition constructs and multiple roles would be beneficial when schools measure bullying involvement at the universal level (i.e., school-wide/Tier 1).

This study found a total of 48 assessment tools, with 20 categorized as measuring both bullying and victimization and 13 categorized as measuring bullying, victimization, and bystander roles. This suggests a variety of measures exist for schools to select, yet as seen in Table 4 and Table 5, some of these assessment tools do not address all three definition components (e.g., Modified Peer Nomination Inventory [58], Cyber-Harassment Student Survey [69]). Further, each assessment tool focuses on a different combination of types of bullying. As such, schools should select an assessment tool that measures all three constructs of the definition, as well as the types of bullying involvement of interest. For example, if a school is concerned with both perpetrators and victims of cyberbullying, they might select the Cyberbullying and Online Aggression Survey [59] or the European Cyberbullying Intervention Project Questionnaire [67], which appropriately addresses all areas of concern. Alternatively, if a school did not have a specific area of concern but wanted to administer an assessment to get an overview of bullying behaviors, they might select an assessment tool addressing multiple types and roles (e.g., Bully Survey [79], The Colorado Trust Bullying Prevention Initiative Student Survey [73]).

This study found five assessment tools that addressed bullying only and 12 assessment tools that addressed victimization only. Currently, some schools are moving to implement Multi-Tiered Systems of Support (MTSS) for bullying prevention. In the MTSS system, a comprehensive assessment would be necessary at Tier 1 (e.g., an assessment tool addressing multiple roles and types of bullying), but in Tiers 2 and 3, a targeted assessment would be the best to understand students who may require more intensive interventions. Assessment tools identified as bully-only (Table 2) or victim-only (Table 3) are satisfactory for a targeted population, as they do not comprehensively measure bullying roles. To illustrate, a school might elect to administer the Gatehouse Bullying Scale to a small group of students they suspect are victims to gauge the students’ experience with physical, verbal, and relational bullying and further inform student support. It is important to note that some assessments included in this study are specific to one type of bullying, for example, the Weight-Based Teasing Scale [40] or the Gay, Lesbian, Straight, Education Network (GLSEN) National School Climate Survey [44]. These assessment tools could be selected to assess specific biases or victim experiences.

Finally, this study analyzed types of bullying measured by individual items. Out of the total items included for analysis, verbal bullying was measured most frequently, followed by relational and physical bullying, and cyberbullying was measured by only about 8% of the total items. A recent meta-analysis examining bullying patterns indicates that trends differ based on the type of bullying [82]. Specifically, over the last two decades, there has been a notable decrease in both physical and verbal bullying victimization, whereas cyberbullying has shown an increase [82]. With the recent rise in cyberbullying rates, schools are increasingly looking to prevent and intervene. Inconsistent measurement strategies can also increase the difficulty in monitoring the problem and evaluating the impact and progress of bullying prevention interventions [27]. It is recommended that schools choose an assessment tool that measures all three constructs of the bullying definition, with a combination of roles and types appropriately addressing all areas of concern to best inform intervention.

### 4.1. Limitations

Limitations associated with this study should be noted. First, the literature review did not closely adhere to the recommended guidelines by PRISMA to be considered a systematic review [83]. However, this study provided a content analysis of available assessment tools concerning the definition of bullying that resulted from a thorough review of the literature. Specifically, the researchers conducted an exhaustive literature search to find measures the *Compendium of Assessment Tools* may have missed [22]. Second, assessment tools were only reviewed if they were readily available or provided by the original authors. Therefore, assessment tools only available behind a paywall were not reviewed or considered for the content analysis. Finally, assessments designed to measure tangentially related constructs were not considered, even if they contained items similar to the bullying-specific measures. As such, only assessment tools claiming to measure bullying involvement specifically were considered and retained for the content analysis.

### 4.2. Implications

The results of this study have implications for both researchers and practitioners. Researchers should be very intentional when developing constructs as new assessment tools are being established and validated. If the assessment is intended to measure bullying, victimization, or bully-victims, the definition should be directly measured by each item, including intentionality, power imbalance, and repetition of behaviors. Additionally, if the assessment is designed to measure bullying as an overarching construct, the different typographies of aggressive behaviors should be assessed, including physical, verbal, relational, property damage, and cyber, to capture the most accurate representation of the prevalence of bullying within a school or district. Single-item indicators of bullying involvement are not satisfactory enough to measure the complex nature of bullying [25,26], as they often do not measure multiple behavioral domains or provide a clear representation of defining characteristics. While all of the measures in this study have been evaluated and have acceptable psychometrics, selecting a measure for a particular study hinges on the specific aims and research questions. Specifically, researchers are encouraged to select a measure that most accurately measures the construct of interest. When measuring bullying involvement, scholars are encouraged to consider a measure that best represents the type of bullying of interest, the role of the respondent (i.e., perpetrator, victim, bully-victim, bystander), and items or constructs that include the defining characteristics.

For practitioners, this study suggests being intentional when selecting an assessment. While there are many tools designed to measure bullying involvement of school-aged youth, and there are specific tools developed to measure various aspects of bullying involvement (e.g., weight-based bullying), it is recommended that schools choose an assessment that fits their needs. For example, schools should consider their unique needs and develop a climate assessment, including a bullying involvement measure, that evaluates those needs. When selecting a measure, in addition to assessing the unique needs of a given school or district, school officials should determine how often to assess (2–3 times per year), primary respondents (e.g., students, teachers, parents), respondent role, types of bullying to measure, the degree to which the measure assess defining characteristics, other protective and predictive factors in assessing (e.g., school belonging, empathy), and time allotment or number of items to include [10,84,85]. Given the necessity to accurately measure bullying involvement, it is recommended that a school develop a bullying prevention team or task force to aid in the instrument selection and data interpretation [86,87,88]. Given the findings of the current study, several validated measures will meet the individual needs of most K-12 educational environments.

## 5. Conclusions

In conclusion, this item-level analysis comprehensively examined the frequencies and percentages across various dimensions of bullying assessments. The findings revealed a diverse landscape relative to the measurement and assessment of bullying, victimization, and bystander involvement among school-aged youth. Additionally, this study revealed distinct patterns related to various assessment domains. Notably, the bully-only assessments, which exclusively focused on bullying perpetration, demonstrated a predominant emphasis on physical and verbal bullying perpetration. Victim-only assessments, which prioritized the measurement of victimization among school-aged youth, had an emphasis on the prevalence of verbal and relational victimization. The bully and victim assessments demonstrated a balance between perpetration and victimization while emphasizing intent and power imbalance across assessments. Finally, the bully, victim, and bystander assessments highlighted the complexity of addressing all three roles, with variations in the typographies of bullying measured and the extent to which power imbalance was represented. While it should be noted that all measures in this study have been validated and reported to have acceptable psychometric properties, selecting an instrument hinges on the overarching purpose of measurement. Specifically, scholars should select a measure that most accurately evaluates the study’s construct(s) of interest, while practitioners should select a measure that most accurately provides data related to the unique needs of an individual school or district. Furthermore, these findings underscore the need for more nuanced and comprehensive assessments to capture the multifaceted and complex nature of bullying involvement among school-aged youth.

## Figures and Tables

**Table 1 ijerph-22-00029-t001:** Frequencies and percentages for item-level analysis (*n* = 1499).

Domain	*n*	%
Roles		
Bully	318	21
Victim	577	38
Bystander: Outsiders	148	10
Bystander: Kids who Reinforce	10	<1
Bystander: Kids who Defend	36	2
Bystander: Kids who Assist	12	<1
None	398	27
Type		
Sexual	100	7
Physical	200	13
Verbal	255	17
Relational	208	14
Cyberbullying	123	8
Damage to Property	39	3
Illegal	51	3
None	437	29
Definition Constructs		
Repetition	800	53
Intent	874	58
Power Imbalance	375	25
None	568	38

**Table 2 ijerph-22-00029-t002:** Bully-only assessments.

Citation	Scale	Type	Definition Components
Bosworth et al. (1999) [29]	Modified Aggression Scale	G, P, V, R	REP, I, PI
Bryant (1993) [38]	AAUW Sexual Harassment Survey	S, V, R	REP, I, PI
Farmer & Aman (2009) [15]	Children’s Scale of Hostility and Aggression: Reactive/Proactive (C-SHARP)—Parent	G, S, P, V, R, D	REP, I, PI
Goodman et al. (1998) [39]	Strengths and Difficulties Questionnaire	G, P	I
Orpinas & Frankowski (2001) [28]	Aggression Scale	P, V, R	REP, I, PI

Note. rD = Damage to Property, G = General, I = Intent, P = Physical, PI = Power Imbalance, R = Relational, REP = Repetition, S = Sexual, V = Verbal.

**Table 3 ijerph-22-00029-t003:** Victim-only assessments.

Citation	Scale	Types	Definition Components
Arora (1994) [40]	“My Life in School” Checklist	G, P, V, R, D	REP, I, PI
Bond et al. (2007) [41]	Gatehouse Bullying Scale	P, V, R	REP, I, PI
Eisenberg et al. (2003) [42]	Weight-Based Teasing Scale	V	REP, I, PI
Green et al. (2018) [43]	California Bullying Victimization Scale	S, P, V, R, D	REP, I, PI
Hall (2016) [44]	The Bullying, Harassment, and Aggression Receipt Measure (Bullyharm)	S, P, V, R, C, D	REP, I, PI
Hunt et al. (2012) [45]	Personal Experiences Checklist (PECK)	P, V, R, C, D	REP, I, PI
Kosciw & Diaz (2008) [46]	Gay, Lesbian, Straight, Education Network (GLSEN) National School Climate Survey	G, S, P, V, R, C, D	REP, I, PI
Morton et al. (2021) [47]	The Assessment of Bullying Experiences—Parent	P, V, R, C, D	REP, I, PI
Mynard & Joseph (2000) [48]	Multidimensional Peer-Victimization Scale	P, V, R, D	REP, I, PI
Orpinas (1993) [49]	Victimization Scale	P, V, R	REP, I, PI
Strout et al. (2018) [50]	Child Adolescent Bullying Scale (CABS)	G, P, V, R, C, D	REP, I, PI
Thompson et al. (1995) [51]	Perception of Teasing Scale (POTS)	V	REP, I, PI

Note. C = Cyber, D = Damage to Property, G = General, I = Intent, P = Physical, PI = Power Imbalance, R = Relational, REP = Repetition, S = Sexual, V = Verbal.

**Table 4 ijerph-22-00029-t004:** Bully and victim assessments.

Citation	Scale	Types	Definition Components
Chan et al. (2005) [52]	School Life/Survey	P, V, R, C, D	REP, I, PI
Crick & Grotpeter (1995) [53]	Relational Aggression and Victimization Scales	R	REP, I, PI
Espelage & Holt (2001) [30]	Illinois Bully Scale	G, P, V, R	REP, I, PI
Gottheil & Dubow (2001) [54]	Setting the Record Straight	G, P, V	I
Gotthiel & Dubow 2001) [55]	Introducing My Classmates	P, V	I
Murray et al. (2021) [56]	Zurich Brief Bullying Scale	S, P, V, R, D	REP, I, PI
Orpinas & Horne (2006) [57]	Reduced Aggression/Victimization Scale	P, V, R	REP, I, PI
Parada (2000) [58]	Adolescent Peer Relations Instrument	P, V, R, D	REP, I, PI
Patchin & Hinduja (2006) [59]	Cyberbullying and Online Aggression Survey	C	REP, I, PI
Perry et al. (1988) [60]	Modified Peer Nomination Inventory	G, P, V	I
Poteat & Espelage (2005) [61]	Homophobic Content Agent Target Scale	V	REP, I, PI
Roberson & Renshaw (2018) [62]	Health Behavior School-Aged Children Survey	S, P, V, R, C	REP, I, PI
Saylor et al. (2012) [63]	The Bullying and Ostracism Screening Scales (BOSS)	P, V, R, C	REP, I, PI
Shaw et al. (2013) [64]	Forms of Bullying Scale—Victimization version (FBS-V) Forms of Bullying Scale—Perpetration version (FBS-P)	P, V, R, C	REP, I, PI
Tarshis & Huffman (2007) [65]	Peer Interactions in Primary School Questionnaire	G, P, V, R, D	REP, I, PI
Warden et al. (2003) [66]	Child Social Behavior Questionnaire	P, V, R, D	REP, I, PI
Williford & DePaolis (2019) [67]	European Cyberbullying Intervention Project Questionnaire	C	REP, I, PI
Wolke et al. (2000) [68]	School Relationships Questionnaire	G, P, V, R, D	REP, I, PI

Note. C = Cyber, D = Damage to Property, G = General, I = Intent, P = Physical, PI = Power Imbalance, R = Relational, REP = Repetition, S = Sexual, V = Verbal.

**Table 5 ijerph-22-00029-t005:** Bystander, bully, and/or victim scales.

Citation	Scale	Roles	Types	Definition Components
Austin & Joseph (1996) [23]	Peer Victimization Scale	BY	G, P, V, R	I, PI
Brean & Li (2005) [69]	Cyber-Harassment Student Survey	B, VM, BY	G, C	R, I
Björkqvist & Österman (1995) [70]	Peer Estimated Conflict Behavior Inventory	B, VM, BY	P, V, R	REP, I, PI
Bochaver et al. (2019) [71]	The School Bullying Risk Survey	BY	G, P, V	I, PI
Brown et al. (2011) [72]	Teacher Assessment of Student Behavior	B, BY	P, V, R, C	REP, I, PI
Csuti (2008) [73]	The Colorado Trust Bullying Prevention Initiative Student Survey	B, VM, BY	P, V, R, C	REP, I, PI
Espelage et al. (2012) [74]	Willingness to Intervene	BY	V	R
Fitzpatrick & Bussey (2011) [75]	Social Bullying Involvement Scales	B, VM, BY	R	REP, I, PI
Nadel et al. (1996) [76]	Exposure to Violence and Violent Behavior Checklist	B, VM, BY	S, P, V, IL	REP, I, PI
Salmivalli et al. (2004) [77]	Participant Role Questionnaire	B, BY	G	REP, I, PI
Schäfer et al. (2004) [78]	Retrospective Bullying Questionnaire	B, VM, BY	G, P, V, R, D	REP, I, PI
Swearer & Cary (2003) [79]	Bully Survey	B, VM, BY	G, P, V, R, D	REP, I, PI

Note. B = Bullying, BY = Bystander, C = Cyber, D = Damage to Property, G = General, I = Intent, IL = Illegal, P = Physical, PI = Power Imbalance, R = Relational, REP = Repetition, S = Sexual, V = Verbal, VM = Victim.

## Data Availability

The data for this study are available with a request to the author.

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
