# Peer review of "Literature Review and Content Analysis of Bullying Assessments: Are We Measuring What We Intend to?"

_ijerph, 2024, doi:10.3390/ijerph22010029_

Round 1

Reviewer 1 Report

Comments and Suggestions for Authors

The topic addressed in the article is highly relevant, especially due to the significant variations in bullying percentages. It is crucial to clarify certain aspects to improve the content. Below are my observations and suggestions to make the manuscript suitable for publication.

Introduction:

The introduction is well-structured and supported by a wide range of references. However, I believe it is necessary to specify, when describing the main questionnaires used, when each one considers a behavior as bullying. Some questionnaires define bullying as behavior repeated once a week, while others establish it as twice a month. It would be helpful to clarify this variation in the referenced citations.

Materials and Methods:

The methodological approach is adequate, though it would have been beneficial to expand the search to broader databases like WoS and Scopus, where more impactful articles can be found. Additionally, the PRISMA protocol, which is essential for this type of systematic review, has not been followed.

Results:

The results presented are of great interest to the scientific community. However, the tables do not follow the correct format and are difficult to understand. If this aspect is not corrected, the article cannot be published. It is essential that the results be presented more clearly and in a visually accessible way, as the information is currently dense and hard to read, which limits the feedback I can provide.

Discussion:

Instead of repeating the results, the discussion should focus on which evaluation tool would be the most appropriate to use. A more comparative approach is needed between the different tools evaluated, rather than simply describing each one.

Limitations:

Although the PRISMA protocol was not used, the information provided in the article is valuable. However, it is important to mention this limitation clearly in the article itself.

Conclusions:

The conclusions and practical implications should be more concise. It is important to clearly address the study objectives and indicate which tools would be most useful for each evaluated role, based on the review. Including a summary table would help clarify these points.

Formatting:

The article does not comply with the format required by the journal, which makes it considerably harder to read. The authors should adjust the manuscript to the requested format and add line numbers to facilitate a more thorough and specific review.

Author Response

Comment 1: The introduction is well-structured and supported by a wide range of references. However, I believe it is necessary to specify, when describing the main questionnaires used, when each one considers a behavior as bullying. Some questionnaires define bullying as behavior repeated once a week, while others establish it as twice a month. It would be helpful to clarify this variation in the referenced citations.

Response 1: Thank you for this comment. We have clarified this in the introduction. Information has been added to the introduction to describe inconsistencies in measurement between scales.

Comment 2: The methodological approach is adequate, though it would have been beneficial to expand the search to broader databases like WoS and Scopus, where more impactful articles can be found. Additionally, the PRISMA protocol, which is essential for this type of systematic review, has not been followed.

Response 2: Thank you for these recommendations. In the future our team will be sure to include broader databases such as WoS and Scopus. Given the primary goal of our literature search was to identify assessment tools in addition to those presented in Hamburger et al., (2011) Measuring Bullying Victimization, Perpetration, and Bystander Experiences: A Compendium of Assessment Tools we elected to conduct our search without strictly adhering to the PRISMA protocol. The decision to employ a discontinuation criterion came from the purposefully broad search terms selected paired with the large number of initial results. We have added a statement to the limitations section regarding the PRISMA standards.

Comment 3: The results presented are of great interest to the scientific community. However, the tables do not follow the correct format and are difficult to understand. If this aspect is not corrected, the article cannot be published. It is essential that the results be presented more clearly and in a visually accessible way, as the information is currently dense and hard to read, which limits the feedback I can provide.

Response 3: Thank you for your feedback. The tables have all been revised to the correct format for readers to better understand the results of the paper.

Comment 4: Instead of repeating the results, the discussion should focus on which evaluation tool would be the most appropriate to use. A more comparative approach is needed between the different tools evaluated, rather than simply describing each one.

Response 4: Thank you for pointing out this concern. We have updated this section to include a more in depth discussion of when or why schools might elect to administer different assessment tools. Additionally, we have made the connection to Multi-Tiered Systems of Support clearer to further support the updated guidance outlined in the discussion.

Comment 5: Although the PRISMA protocol was not used, the information provided in the article is valuable. However, it is important to mention this limitation clearly in the article itself.

Response 5: We have added a statement in the limitations section regarding the PRISMA protocol.

Comment 6: The conclusions and practical implications should be more concise. It is important to clearly address the study objectives and indicate which tools would be most useful for each evaluated role, based on the review. Including a summary table would help clarify these points.

Response 6: hank you for the comment and we recognize the need to be more concise in our implications and conclusion. We have addressed this by providing recommendations for scholars and school-based professionals when selecting an instrument to measure bullying. Given the complexity and range of the instruments evaluated, we did not include a summary table, as selecting an instrument is grounded in the research questions (scholar) or unique needs of a school (practitioner). With that said, we hope that we have directly addressed this comment and provided more clarity in these sections.

Comment 7: The article does not comply with the format required by the journal, which makes it considerably harder to read. The authors should adjust the manuscript to the requested format and add line numbers to facilitate a more thorough and specific review.

Response 7: 

Thank you for your comment. The references have been changed to fit the format of the Journal. References are numbered in their order of appearance in the text (including table captions and figure legends) and listed individually at the end of the manuscript.

Reviewer 2 Report

Comments and Suggestions for Authors

Thank you for the opportunity to review this manuscript. I recommend rejecting this article for publication for several reasons described below that include, but are not limited to:

1. The literature review lacks empirical information to provide readers with a comprehensive understanding of existing knowledge.

2. The methodology section (PRISMA) is underdeveloped and needs more attention to make it a publishable article.

3. The purpose of the study changes throughout the manuscript, making it difficult to understand what the authors are trying to answer.

4. There are many errors in the references in APA, they appear unstructured.

Author Response

Comment 1: The literature review lacks empirical information to provide readers with a comprehensive understanding of existing knowledge.

Response 1: Thank you for this comment. We have added additional empirical information (specifically about the Measuring Bullying Victimization, Perpetration, and Bystander Experiences: A Compendium of Assessment Tools) to provide the reader with more of an understanding of what already exists that contributes to this topic.

Comment 2: The methodology section (PRISMA) is underdeveloped and needs more attention to make it a publishable article.

Response 2: Thank you for pointing out this concern. We have revised the methodology section to read more clearly and have added a statement to the limitations section regarding the PRISMA standards.

Comment 3: The purpose of the study changes throughout the manuscript, making it difficult to understand what the authors are trying to answer.

Response 3: Thank you for this comment. The discussion and conclusions section has been edited to more clearly align with the purpose of the paper identified in the introduction.

Comment 4: There are many errors in the references in APA, they appear unstructured.

Comment 5: 

Thank you for your comment. The references have been changed to fit the format of the Journal. References are numbered in their order of appearance in the text (including table captions and figure legends) and listed individually at the end of the manuscript.

Reviewer 3 Report

Comments and Suggestions for Authors

The abstract does not include a reference, please delete it.

The aim of this study is to examine and analyze abuse assessment tools.

This is extremely useful research, which can have significant implications in practice, that is, it can contribute to the creation of better tools that will more precisely identify violent behavior.

The research questions are well posed.

The chosen research method for the papers is appropriate.

Detailed and understandable data analysis. Authors have successfully identified key assessment tools that enable a better understanding of the phenomenon of bullying, which is crucial for developing effective interventions.

The authors successfully integrate recommendations for further research, demonstrating their commitment to the continuous enhancement of scientific knowledge about youth bullying.

Author Response

Comment 1: The abstract does not include a reference, please delete it.

Response 1: This has been deleted.

Comment 2: 

This is extremely useful research, which can have significant implications in practice, that is, it can contribute to the creation of better tools that will more precisely identify violent behavior.

The research questions are well posed.

The chosen research method for the papers is appropriate.

Detailed and understandable data analysis. Authors have successfully identified key assessment tools that enable a better understanding of the phenomenon of bullying, which is crucial for developing effective interventions.

The authors successfully integrate recommendations for further research, demonstrating their commitment to the continuous enhancement of scientific knowledge about youth bullying.

Response 2: Thank you for your kind words.

Round 2

Reviewer 1 Report

Comments and Suggestions for Authors

The authors have conducted a thorough and detailed review of the topic, adequately addressing the comments and suggestions provided.